# Context-dependent modulations of subthalamo-cortical synchronization during rapid reversals of movement direction in Parkinson's disease

Lucie Winkler[1], Markus Butz[1], Abhinav Sharma[2,3], Jan Vesper[4], Alfons Schnitzler[1], Petra Fischer[5]*, Jan Hirschmann[1]

[1]Institute of Clinical Neuroscience and Medical Psychology, Medical Faculty and University Hospital Düsseldorf, Heinrich Heine University Düsseldorf, Duesseldorf, Germany; [2]Medical Research Council Brain Network Dynamics Unit, University of Oxford, Oxford, United Kingdom; [3]Nuffield Department of Clinical Neurosciences, University of Oxford, Oxford, United Kingdom; [4]Department of Functional Neurosurgery and Stereotaxy, Medical Faculty and University Hospital Düsseldorf, Düsseldorf, Germany; [5]School of Physiology, Pharmacology & Neuroscience, University of Bristol, Bristol, United Kingdom

*For correspondence:
petra.fischer@bristol.ac.uk

Competing interest: The authors declare that no competing interests exist.

## eLife Assessment

This **valuable** study combined whole-head magnetoencephalography (MEG) and subthalamic (STN) local field potential (LFP) recordings in patients with Parkinson's disease undergoing deep brain stimulation surgery. The paper provides **convincing** evidence that cortical and STN beta oscillations are sensitive to movement context.

**Abstract** The role of beta band activity in cortico-basal ganglia interactions during motor control has been studied extensively in resting-state and for simple movements, such as button pressing. However, little is known about how beta oscillations change and interact in more complex situations involving rapid changes of movement in various contexts. To close this knowledge gap, we combined magnetoencephalography (MEG) and local field potential recordings from the subthalamic nucleus (STN) in Parkinson's disease patients to study beta dynamics during initiation, stopping, and rapid reversal of rotational movements. The action prompts were manipulated to be predictable vs. unpredictable. We observed movement-related beta suppression at motor sequence start, and a beta rebound after motor sequence stop in STN power, motor cortical power, and STN-cortex coherence. Despite involving a brief stop of movement, no clear rebound was observed during reversals of turning direction. At the cortical level, beta power decreased bilaterally following reversals, but more so in the hemisphere ipsilateral to movement, due to a floor effect on the contralateral side. In the STN, power modulations varied across patients, with patients displaying brief increases or decreases of high-beta power. Importantly, cue predictability affected these modulations. Event-related increases of STN-cortex beta coherence were generally stronger in the unpredictable than in the predictable condition. In summary, this study reveals the influence of movement context on beta oscillations in basal ganglia-cortex loops when humans change ongoing movements according to external cues. We find that movement scenarios requiring higher levels of caution involve enhanced modulations of subthalamo-cortical beta synchronization. Furthermore, our results confirm that beta oscillations reflect the start and end of motor sequences better than movement changes within a sequence.

## Introduction

Beta oscillations within cortical sensorimotor areas and the basal ganglia have been proposed to play a role in movement initiation, termination, and inhibition (**Benis et al., 2014**; **Jurkiewicz et al., 2006**; **Wessel, 2020**). Altered beta band activity has been strongly linked to motor impairment in Parkinson's disease (PD), demonstrating its relevance to proper motor performance (**Brown et al., 2001**; **Cassidy et al., 2002**; **Tinkhauser et al., 2017**). The beta rhythm is often interpreted as reflecting the status quo (**Engel and Fries, 2010**), that is, active maintenance or stabilization of current motor or cognitive output to attenuate alternatives and distractions (**Espenhahn et al., 2017**; **Fischer et al., 2019**). The basal ganglia keep cortex under inhibitory control (**Bonnevie and Zaghloul, 2019**) which, similarly to releasing a break in a car, must be removed to change the current motor state (**Alegre et al., 2013**).

Starting and stopping of movement have mostly been studied with variations of the Stop Signal Task and the Go/No-Go Task (**Alegre et al., 2013**; **Aron and Poldrack, 2006**; **Ray et al., 2012**) that require participants to perform simple, ballistic movements and inhibit them occasionally. Shortly before and during movement, beta oscillations are suppressed (beta suppression), reflecting a task-related active state of the motor network (**Jurkiewicz et al., 2006**). In contrast, beta power transiently increases above baseline levels after movement termination (beta rebound) (**Fonken et al., 2016**; **Ray et al., 2012**), indicating inhibition (**Salmelin et al., 1995**; **Schmidt and Berke, 2017**) and motor adaptation processes (**Struber et al., 2021**; **Tan et al., 2014**). Whether these modulations are causally involved in motor control is still under debate (**Pfurtscheller et al., 2005**; **Toledo et al., 2016**).

Response inhibition has been associated with increased beta power or reduced suppression thereof in prefrontal cortical areas (**Swann et al., 2009**; **Wagner et al., 2018**), and in the subthalamic nucleus (STN) (**Bastin et al., 2014**), with some studies reporting correlations with inhibitory success (**Benis et al., 2014**; **Chen et al., 2020**). Besides playing a critical role in stopping movement (**Mosher et al., 2021**), the STN seems to be involved in delaying or pausing movement until sufficient evidence in favor of a motor program has accumulated (**Ray et al., 2012**). Recent evidence demonstrated that the STN is recruited by the prefrontal cortex via the hyperdirect pathway to implement its pausing function (**Chen et al., 2020**; **Lofredi et al., 2021**; **Oswal et al., 2021**; **Wessel et al., 2019**). However, the role of cortico-subcortical beta synchronization in coordinating movements that are already ongoing remains elusive.

Communication between STN and cortex might become particularly important in tasks involving cognitive factors, such as anticipation. STN beta power has been found to index task complexity and behavioral control (**Oswal et al., 2013**), proactive inhibition and planning (**Benis et al., 2014**), non-motor decision making, and working memory (**Zavala et al., 2017**), and cue evaluation with respect to behavioral goals (**Oswal et al., 2012**). Yet, the extent to which modulation of beta oscillations in basal-ganglia cortex networks depends on expectation remains unknown.

In the present study, we address these research gaps with a paradigm that involves a rotational movement performed in a continuous fashion with occasional rapid changes in movement direction (reversals), as well as movement initiations and terminations. Accounting for the relevance of basal ganglia-cortical loops in motor control, we recorded cortical and STN oscillatory activity simultaneously in PD patients who had undergone implantation of deep brain stimulation (DBS) electrodes the day before. Patients performed the rotational movements according to visual instructions which were manipulated such that their identity and time of appearance was either predictable or unpredictable. With this design, we aimed (1) to investigate the dynamics of STN and STN-cortex beta synchronization during movement reversal compared to those of starting and stopping and (2) to assess the effect of the temporal predictability of movement instructions on the coordination of beta synchronization for starting, stopping, and reversing.

## Results

### Behavior

Patients turned a wheel (**Figure 1b**) with their index finger at their preferred speed and were prompted by visual cues to start, reverse, or stop rotational movement (**Figure 1a**) while we simultaneously measured MEG and STN local field potentials (LFPs). We considered an average of 60.1 (SD = 14.8) predictable start trials, 59.3 (SD = 17.8) unpredictable start trials, 59.9 (SD = 14.2) predictable reversal trials, 58.6 (SD = 15.5) unpredictable reversal trials, 61.3 (SD = 15.3) predictable stop trials and 60.2

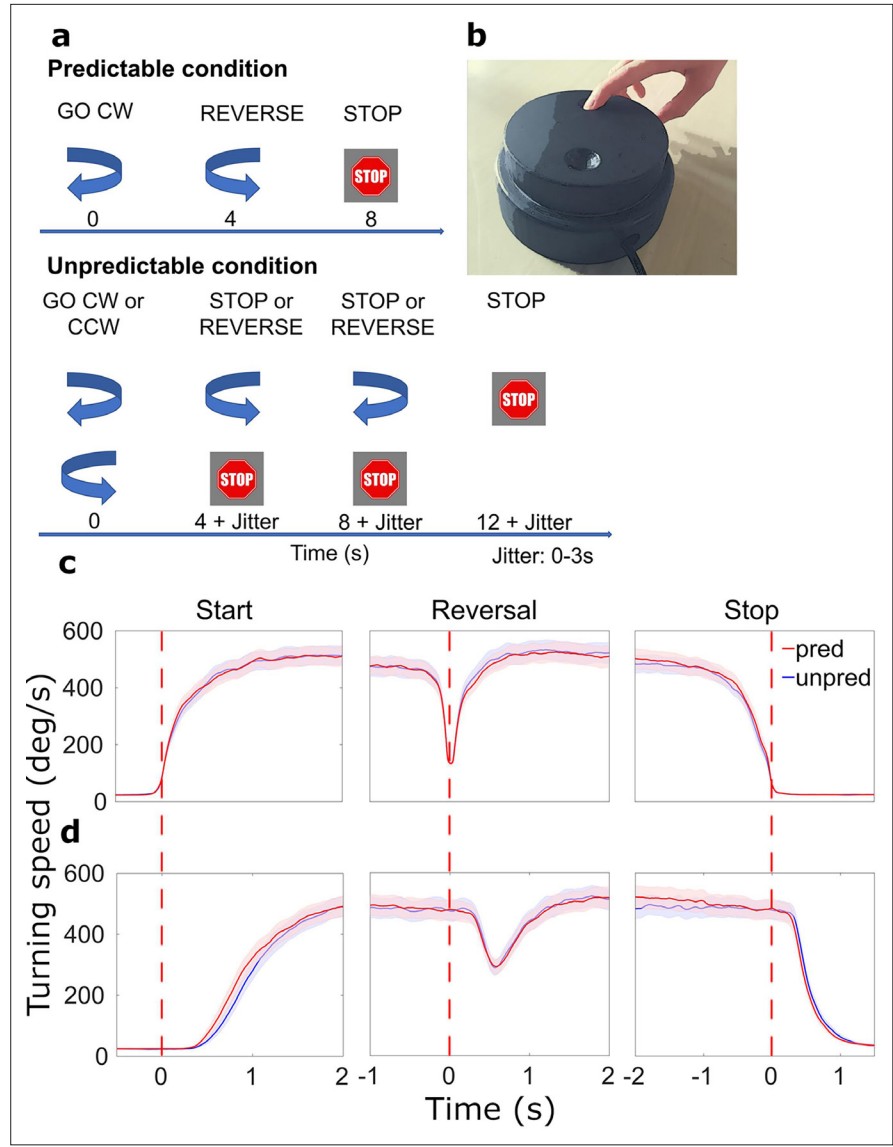

**Figure 1.** Paradigm and behavioral results. (**a**) Patients were cued by arrows to start turning or reverse movement direction. Stop cues were presented at the end of each sequence. The timing of cues varied with the condition: in the predictable condition, the start cue was always followed by a reverse cue after 4 s and a stop cue after another 4 s (no jitter). In the unpredictable condition, there were either 0, 1, or 2 reversals (equal probability). Cue onset was jittered. CW: clockwise, CCW: counterclockwise. (**b**) Turning device for motor paradigm. (**c**) Average movement-aligned wheel speed. Red dotted lines indicate when turning began, was reversed in direction, and halted. (**d**) Average cue-aligned wheel speed. Red dotted lines indicate when the start, reversal, and stop cues appeared, respectively. N=20.

(SD = 17.4) unpredictable stop trials per patient for analysis. To assess whether the predictability of movement prompts had an effect on behavior, we analyzed its effect on movement speed and reaction times. Angular speed changes in start, reversal, and stop trials were similar in the predictable and the unpredictable condition when the data was aligned to action onset (**Figure 1c**, $F_{cond}(1,16) = 0.037$, $p_{cond} = 0.850$, $\eta p^2=0.002$; see **Supplementary file 1** for the complete results of the ANOVA). Aligning trials to cue onsets revealed that starting and stopping occurred slightly later in the unpredictable condition (**Figure 1d**). This was reflected by a main effect of *condition* ($F_{cond}(1,16) = 6.698$, $p_{cond} = 0.020$, $\eta p^2=0.295$) and a *condition\*movement type* interaction effect ($F_{cond*mov}(2,15)=4.916$, $p_{cond*mov} = 0.023$, $\eta p^2=0.396$) on reaction times. Post-hoc *t*-tests revealed that reaction times to predictable start cues (M=0.757, SD = 0.154) and stop cues (M=0.824, SD = 0.202) were significantly shorter

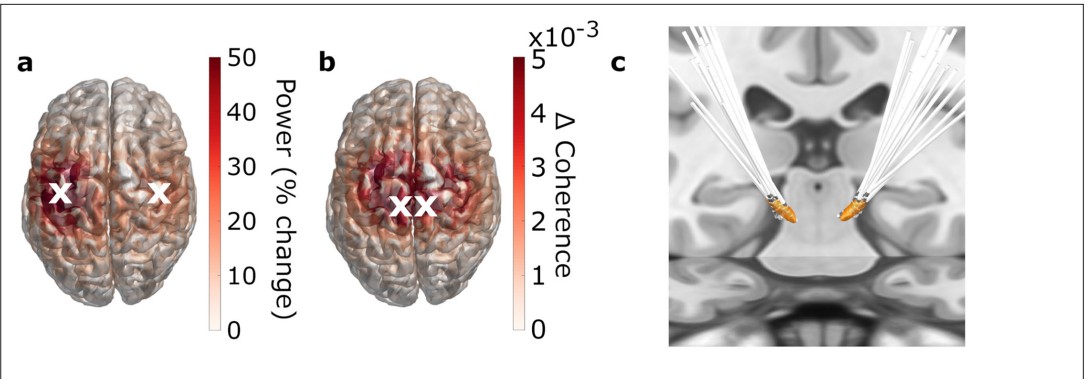

**Figure 2.** Regions of interest. (**a, b**) 3D source-reconstruction in MNI space. N=20. White crosses mark the cortical regions of interests (ROIs) selected for further analysis based on the strongest relative change in power (**a**) and the strongest absolute change in coherence (**b**). (**c**) All patients' deep brain stimulation (DBS) electrodes, localized with Lead-DBS.

than reaction times to unpredictable start (*M*=0.840, SD = 0.160) and stop (*M*=.889, SD = 0.233) cues (start: *t*=–3.469, one-sided p=0.001, *d*=–0.776, stop: *t*=–2.213, one-sided p=0.020, *d*=–0.495). Thus, starting and stopping were not performed at different speeds across conditions, but were initialized earlier in the predictable condition.

## Power

### Modulations of beta power associated with starting and stopping

In order to assess whether beta power modulations associated with reversals were distinct from beta suppression and rebound (research aim 1), we centered the trials on movement initiation, reversal, and termination, respectively, and assessed beta power dynamics. Besides the STN, this was done for two motor cortical regions of interest (ROIs): primary motor cortex (M1, hand knob region) and medial sensorimotor cortex (MSMC). This choice was based on the strongest movement-related modulations of beta power and coherence (*Figure 2*, see Regions of interest in the Methods section for further detail). For comparison, we also present group average time-frequency spectra for the gamma frequency band.

As expected, starting to turn the wheel was associated with a prominent beta suppression (contralateral STN: $t_{clustersum}$ = –2128.9, p<0.001; ipsilateral STN: $t_{clustersum}$ = –2062.8, p<0.001; contralateral M1: $t_{clustersum}$ = –8434.8, p<0.001, ipsilateral M1: $t_{clustersum}$ = –8199.5, p<0.001), whereas stopping resulted in a beta rebound (contralateral STN: $t_{clustersum}$ = 2843.0, p<0.001; ipsilateral STN: $t_{clustersum}$ = 1488.8, p=0.003; contralateral M1: $t_{clustersum}$ = 5958.9, p<0.001, ipsilateral M1: $t_{clustersum}$ = 3834.7, p<0.001) in both motor cortex and STN (*Figures 3a, 4a and b*). The beta suppression occurred bilaterally while the beta rebound was more lateralized to the hemisphere contralateral to movement, as corroborated by a statistical analysis of the lateralization index (*Figure 4c* and *Supplementary file 2*). Power changes in MSMC were generally similar to those in M1.

### Modulations of gamma power associated with starting and stopping

Significant increases in gamma power at movement start were only observed in the contralateral STN ($t_{clustersum}$ = 2216.5, p=0.003, *Figure 3a*). At movement stop, there was a decrease in gamma power in contralateral STN ($t_{clustersum}$ = –734.8, p=0.016, *Figure 3a*) and contralateral M1 ($t_{clustersum}$ = –1447.4, p=0.002, *Figure 4b*). However, changes in gamma power were overall much smaller in magnitude compared to the beta suppression and rebound.

### Modulation of STN beta power associated with reversals

Modulations of beta oscillations associated with reversals of movement direction were of particular interest to this study (research aim 1). When reversing, one first needs to stop the ongoing movement before accelerating again in the opposite direction. Stopping is followed by the beta rebound, whereas starting is preceded by beta suppression. To the best of our knowledge, no study has investigated the

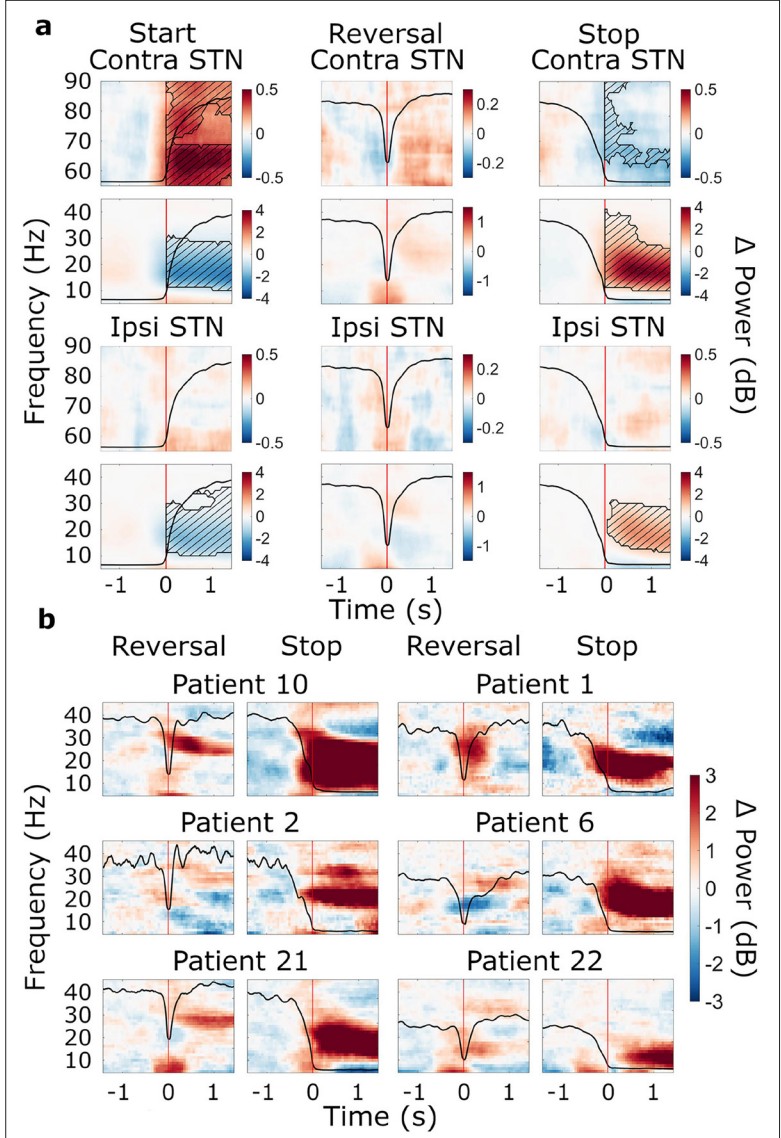

**Figure 3.** Movement-related beta power modulations in the subthalamic nucleus (STN). (**a**) Time-frequency spectra of start, reversal, and stop trials for the STN (group average, trials averaged across predictability conditions). Time 0 marks the moment turning began, was reversed in direction, and halted (red lines). The black line in each plot represents the average wheel turning speed (scale: 0–600 deg/s). Power was baseline-corrected (baseline: –1.6–0 s). Hatched lines within black contours indicate significant changes relative to baseline. N=20. (**b**) Six examples of individual patients at reversal and stop. Power was baseline-corrected (baseline: –1.6–0 s). Time 0 marks the brief pause of movement occurring during reversals, and movement stop, respectively (red lines). The black line in each plot represents each patient's trial-average wheel turning speed (scale: 0–600 deg/s; for patient 21, the scale was adapted to 0–750 deg/s). Patient 10: contralateral, predictable; Patient 1: contralateral, unpredictable, Patient 2: contralateral, predictable; Patient 6: ipsilateral, unpredictable; Patient 21: contralateral, unpredictable; Patient 22; ipsilateral, unpredictable.

The online version of this article includes the following figure supplement(s) for figure 3:

**Figure supplement 1.** Cue-aligned beta power modulations in the subthalamic nucleus (STN).

---

neural signals underlying acceleration that immediately follows stopping. In the STN, reversals were associated with a brief modulation of beta power, which was weak in the group-average spectrum and did not reach significance (*Figure 3a*). Reversal-related beta power modulations of individual patients were variable. Some patients revealed brief increases, whereas others showed decreases in STN beta power upon reaching the turning point (*Figure 3b*). Reversal-related increases of beta power differed

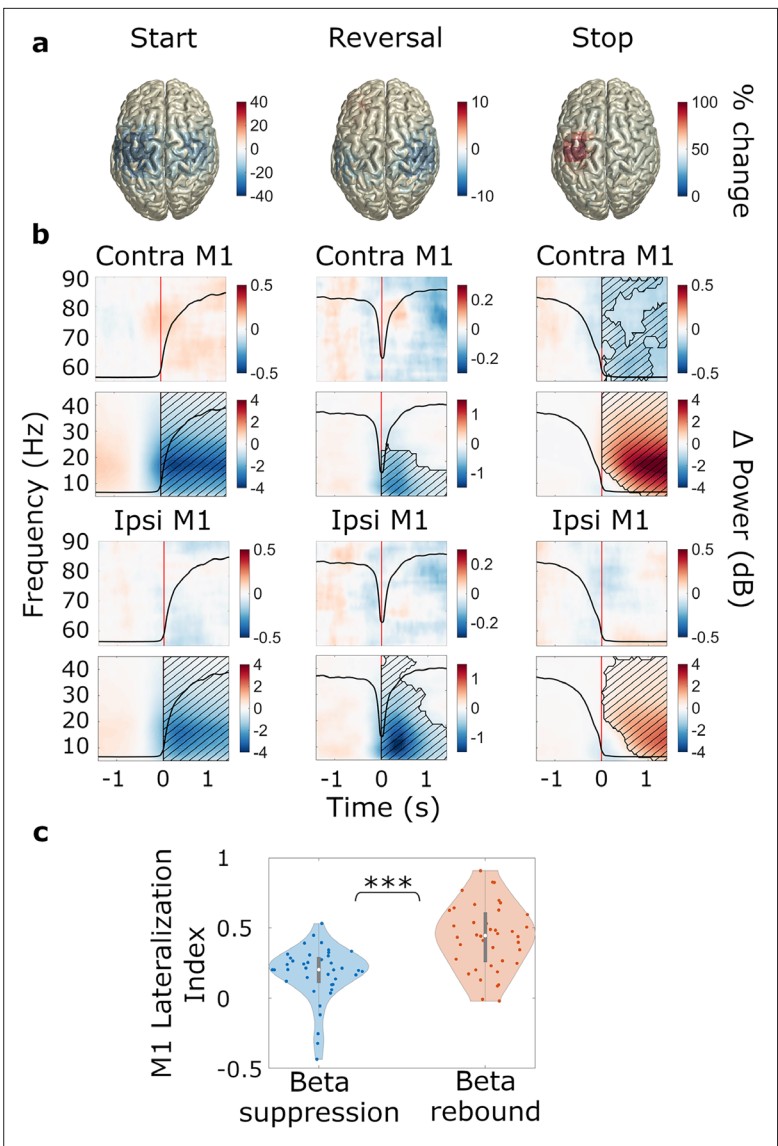

**Figure 4.** Movement-related beta power modulations in M1. N=20. (**a**) Source-localized movement-related modulation of beta power at movement start, reversal, and stop (Montreal Neurological Institute, MNI space, group average, trials averaged across predictability conditions). The hemisphere contralateral to movement is on the left. (**b**) Time-frequency spectra of start, reversal, and stop trials for M1. Time 0 marks the time point turning began, was reversed in direction, and halted (red lines). The black line in each plot represents the average wheel turning speed (scale: 0–600 deg/s). Power was baseline-corrected (baseline: –1.6–0 s). Hatched lines within black contours indicate significant changes relative to baseline. (**c**) Lateralization index for M1. LI = 0 corresponds to no lateralization; positive values refer to a contralateral lateralization and negative values to an ipsilateral lateralization. Blue: beta suppression; red: beta rebound.

The online version of this article includes the following figure supplement(s) for figure 4:

**Figure supplement 1.** Cue-aligned beta power modulations in M1.

from the beta rebound, as occurring after termination of the movement sequence, with respect to amplitude and spectral content, often lacking the low-beta component of the beta rebound (*Figure 3b*). These findings demonstrate distinct processing of brief pauses of action vs. a complete halt of action.

## Modulation of cortical beta power associated with reversals

With respect to cortical beta power dynamics during reversals (research aim 1), we observed that reversals were associated with a brief suppression of alpha and beta power in motor cortex, particularly

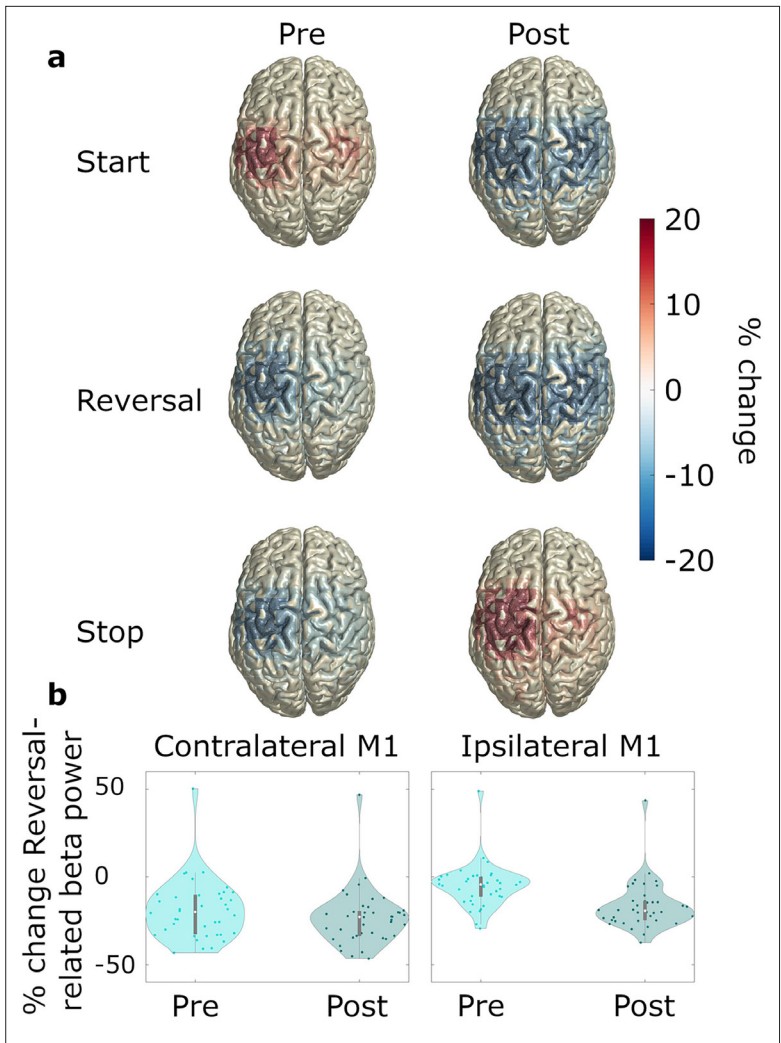

**Figure 5.** Pre- and post-event beta power. *N=20*. (**a**) Source-localized modulation of beta power before and after movement start, reversal, and stop (−1–0 and 0–1 s with respect to the movement of interest; baseline: power averaged over all time points and movement types). Plots are group-averages in Montreal Neurological Institute (MNI) space, trials were averaged across predictability conditions. (**b**) Relative change with respect to whole recording average baseline, of ipsilateral and contralateral beta power for pre-reversal and post-reversal time windows.

in M1 (contralateral M1: $t_{clustersum}$ = −1492.7, p<0.001; ipsilateral M1: $t_{clustersum}$ = −3326.2, p<0.001; *Figure 4a and b*). The suppression occurred after the turning point had been reached (*Figure 4b*) and was stronger in the hemisphere ipsilateral to movement, as demonstrated by a significant *ROI\*movement* interaction effect on baseline-corrected beta power ($F_{ROI*movement}$(10,6)=4.444, $p_{ROI*modulation}$ = 0.041, $\eta p^2$=0.881, refer to *Supplementary file 3* for the full results of the ANOVA). Post-hoc *t*-tests confirmed that beta power was at a lower level in ipsilateral M1 (*M*=−0.047, SD = 0.033) compared to contralateral M1 (*M*=−0.021, SD = 0.023) during reversal of movement direction (*t*=4.454, one-sided p<0.001, *d*=0.996).

To test whether the ipsilateral lateralization was related to pre-event baseline levels (i.e. pre-reversal, pre-start, and pre-stop), we re-computed the modulations using a whole recording average baseline (power averaged over all time points and movement types), thereby omitting the pre- vs. post-event contrast. *Figure 5a* illustrates that movement-related power modulations were generally stronger in the hemisphere contralateral to movement, with the exception of acceleration, which was associated with bilateral suppression of beta power (compare the bilateral beta power suppression at *post-start* and *post-reversal* to the contralateral beta power modulations in all other plots). The second

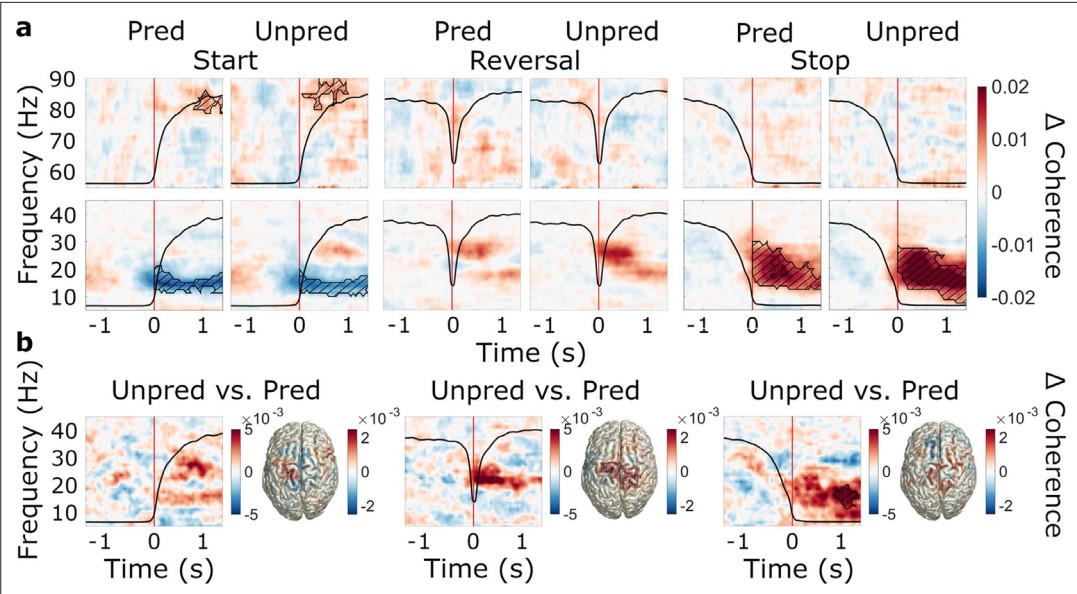

**Figure 6.** Event-related modulations of subthalamic nucleus (STN)-cortex coherence and the effect of predictability. N=20 (**a**) Baseline-corrected group average time-frequency representations of STN-cortex coherence (averaged over regions of interests, ROIs) during start, reversal, and stop for both the predictable and the unpredictable trials (baseline: –1.6–0 s). Time 0 marks the moment turning began, was reversed in direction, and was halted, respectively (red line). The black line in each plot represents the average wheel turning speed (scale: 0–600 deg/s). Hatched lines within black contours indicate significant changes relative to baseline. (**b**) Group average coherence difference between the unpredictable and predictable conditions. Left: Contrast of time-frequency representations. TFRs were averaged over ROIs. Right: Contrast of source-localized, event-related coherence modulations in the beta band.

The online version of this article includes the following figure supplement(s) for figure 6:

**Figure supplement 1.** Cue-aligned modulations of subthalamic nucleus (STN)-cortex coherence and the effect of predictability.

**Figure supplement 2.** Directionality of M1-subthalamic nucleus (STN) and MSMC-STN coupling.

---

before reversing, beta-power was at an intermediate level in the hemisphere ipsilateral to movement (*Figure 5a*). Thus, we observed a suppression relative to the pre-event baseline (*Figure 4a–b*). In the contralateral hemisphere, in contrast, beta power could not be suppressed much further because it was already close to floor level prior to reversing (*Figure 5b*). The lack of a pre-reversal increase of beta power is remarkable, because the second prior to reversing contained the deacceleration of the moving hand, which does not appear to involve an increase of beta power in primary motor cortex.

## Effects of predictability on power

With respect to the effect of predictability of movement instructions on beta power dynamics (research aim 2), we observed an interaction between *movement type* and *condition* ($F_{cond*mov}$ (2,14)=4.206, $p_{cond*mov}$ = 0.037, $\eta p^2$=0.375), such that the beta power suppression at movement start was generally stronger in the predictable (*M*=–0.170, SD = 0.065) than in the unpredictable (*M*=–0.154, SD = 0.070) condition across ROIs (*t*=–1.888, one-sided p=0.037, *d*=–0.422). We did not observe any modulation of gamma power by the predictability of movement instructions ($F_{cond}$ (1,15)=0.792, $p_{cond}$ = 0.388, $\eta p^2$=0.050, **Supplementary file 5**).

## Connectivity

### Movement-related modulations of STN-cortex connectivity

Beyond the local changes in beta power, we intended to investigate the dynamics of oscillatory coupling within the basal ganglia-cortex loop in the context of reversals of movement direction (research aim 1). The movement-related modulations of STN-cortex coherence were similar to modulations of

power, including beta suppression (predictable start: $t_{clustersum}$ = –489.4, p=0.002; unpredictable start: $t_{clustersum}$ = –530.1, p=0.003), beta rebound (predictable stop: $t_{clustersum}$ = 802.0, p=0.002; unpredictable stop: $t_{clustersum}$ = 1252.2, p<0.001), and increases in the gamma band at movement start (predictable start: $t_{clustersum}$ = 120.4, p=0.005; unpredictable start: $t_{clustersum}$ = 197.8, p<0.001). Unlike motor cortical beta power, however, STN-cortex beta coherence did not decrease in the re-acceleration phase of reversals. On a qualitative level, it even increased relative to baseline. *Figure 6* depicts the movement-related changes in coherence averaged across all ROIs.

## Effects of predictability on STN-cortex coherence

With respect to the effect of predictability of movement instructions on beta coherence (research aim 2), we found that the pre-post event differences were generally more positive in the unpredictable condition (main effect of predictability: $F_{cond}(1,15)$ = 8.684, $p_{cond}$ = 0.010, $\eta p^2$=0.367; *Supplementary file 3*), meaning that the suppression following movement start was diminished and the increases following stop and reversal were enhanced in the unpredictable condition (*Figure 6a*). This effect was most pronounced in the MSMC (*Figure 6b*). When comparing region-average TFRs between the unpredictable and the predictable condition, we observed a significant difference only for stopping ($t_{clustersum}$ = 142.8, p=0.023), suggesting that the predictability effect was mostly carried by increased beta coherence following stops. When repeating the rmANCOVA for pre-event coherence, we did not observe an effect of predictability ($F_{cond}(1,15)$ = 0.163, $p_{cond}$ = 0.692, $\eta p^2$=0.011), i.e., the effect was most likely not due to a shift of baseline levels. The increased tendency for upward modulations and decreased tendency for downward modulations rather suggests that the inability to predict the next cue prompted intensified event-related interaction between STN and cortex. STN-cortex gamma coherence was not modulated by predictability ($F_{cond}(1,15)$ = 0.005, $p_{cond}$ = 0.944, $\eta p^2$=0.000, *Supplementary file 5*).

## Granger causality

In general, cortex appeared to drive the STN in the beta band, regardless of the movement type and predictability condition. This was reflected in a main effect of ROI on Granger causality estimates ($F_{ROI}(7,9)$ = 3.443, $p_{ROI}$ = 0.044, $\eta p^2$=0.728; refer to *Supplementary file 4* for the full results of the ANOVA). In the hemisphere contralateral to movement, follow-up *t*-tests revealed significantly greater Granger causality from M1 to the STN (*t*=3.609, one-sided p<0.001, d=0.807) and from MSMC to the STN (*t*=2.051, one-sided p<0.027, d=0.459) than the other way around. The same picture emerged in the hemisphere ipsilateral to movement (M1 to STN: *t*=3.082, one-sided p=0.003, d=0.689; MSMC to STN: *t*=1.833, one-sided p<0.041, d=0.410). In the gamma band, we did not detect a significant drive from one area to the other ($F_{ROI}(7,9)$ = 0.338, $p_{ROI}$ = 0.917, $\eta p^2$=0.208, *Supplementary file 6*). *Figure 6—figure supplement 2* demonstrates the differences in Granger causality between original and time-reversed data for the beta and gamma bands.

# Discussion

Our study demonstrates that initiating, reversing, and stopping a continuous movement involves modulation of local and long-range beta synchronization in basal ganglia-cortex loops. Accelerating, stopping briefly, and coming to a complete halt have distinct and region-specific effects on beta oscillations in the motor system. These effects are context-dependent, with event-related increases of subcortico-cortical coupling intensifying when the upcoming movement instructions cannot be anticipated.

## The dynamics of STN-cortex coherence

Simultaneous measurements of subthalamic and cortical oscillations in a comparably complex motor task allowed us to study the context-dependent dynamics of STN-cortex coupling. STN-M1 and STN-MSMC beta coherence decreased at movement initiation and increased after movement termination, while gamma coherence increased at movement start, corresponding to similar power changes in STN and motor cortex. A pre-movement suppression of beta coherence has been reported previously (*Cassidy et al., 2002*; *Talakoub et al., 2016*; *van Wijk et al., 2017*). Similarly, increases in gamma coherence have been found for the performance of ballistic movements (*Alegre et al., 2013*;

*Litvak et al., 2012*). Stopping a planned movement has been found to be linked with reduced suppression of beta coherence (*Alegre et al., 2013*), but a post-movement increase of coherence has thus far only been described for ballistic movements (*Tan et al., 2014*). Considering the timing of the increase observed here, the STN's role in movement inhibition (*Benis et al., 2014*; *Ray et al., 2012*) and the fact that frontal and prefrontal cortical areas are believed to drive subthalamic beta activity via the hyperdirect pathway (*Chen et al., 2020*; *Oswal et al., 2021*) it seems plausible that the increase of beta coherence reflects feedback of sensorimotor cortex to the STN in the course of post-movement processing. In line with this idea, we observed a cortical drive of subthalamic activity in the beta band.

## Beta coherence and beta power are modulated by predictability

In the present paradigm, patients were presented with cues that were either temporally predictable or unpredictable. We found that unpredictable movement prompts were associated with stronger upward modulations and weaker downward modulations of STN-cortex beta coherence, likely reflecting the patients adopting a more cautious approach, paying greater attention to instructive cues. Enhanced STN-cortex interactions might indicate the recruitment of additional neural resources, which might have allowed patients to maintain the same movement speed in both conditions.

The notion of beta oscillations reflecting motor and cognitive processes such as action selection, clearing, and error-monitoring has gained growing support (*Fonken et al., 2016*; *Schmidt et al., 2019*; *Turner and Desmurget, 2010*). Purely cognitive inhibition processes, such as inhibition of thoughts, have been found to be associated with prefrontal beta power modulations (*Castiglione et al., 2019*; *Schmidt et al., 2019*). Furthermore, the STN has been suggested to implement its *hold your horses* function, reflected by beta-band synchronization, in situations of cognitive conflict (*Brittain et al., 2012*). Simultaneous measurements of MEG and STN LFPs revealed that STN-cortex beta coherence increases after conflict cues in an expanded judgment task (*Patai et al., 2022*). Though the present paradigm did not involve any conflict as such, the context of unpredictable movement instructions possibly engaged similar cognitive processes in response to surprise/uncertainty.

With respect to power, we observed reduced beta suppression in the unpredictable condition at movement start, consistent with the effect on coherence, likely demonstrating a lower level of motor preparation. This finding aligns with MEG research that found reduced beta suppression with enhanced uncertainty in a motor task (*Tzagarakis et al., 2010*), and findings from an EEG study that demonstrated reduced beta suppression in response to an unpredictable sequence of rhythmic stimuli (*Alegre et al., 2003*). Although previous research has reported modulations of the beta rebound by cognitive factors (*Fischer et al., 2016*; *Tan et al., 2016*; *Zavala et al., 2018*), we did not find an effect of predictability on the beta power rebound here.

## Acceleration involves the recruitment of ipsilateral M1

As expected, we found sustained beta suppression at movement start and a strong beta power rebound at movement stop in STN, M1, and MSMC. During reversals, beta power was suppressed briefly in M1, particularly in the ipsilateral hemisphere where beta was not fully desynchronized prior to reversing. In contrast, the contralateral hemisphere revealed a floor effect: the ongoing movement resulted in persistent beta power suppression that was only slightly intensified when reversing. Bilateral modulation of beta power, as reported during reversals, was otherwise observed during the initiation of movement, but not during ongoing movement or after movement termination, suggesting that the recruitment of ipsilateral M1 may be selective to acceleration.

Our findings are consistent with prior studies that have demonstrated a bilateral (*Alegre et al., 2004*; *Zaepffel et al., 2013*) and spatially diffuse (*Jurkiewicz et al., 2006*) beta suppression, and more focal (*Jurkiewicz et al., 2006*) and predominantly contralateral (*Espenhahn et al., 2017*) topography of the beta rebound. Furthermore, past research has posited a role of ipsilateral motor cortex in motor control and preparation (*Jurkiewicz et al., 2006*; *Olson et al., 2022*). Beta suppression in the ipsilateral hemisphere has been found to be related to increased corticospinal excitability, to facilitate finger movements (*Rau et al., 2003*), and to have a role in higher order cortical processing of fine motor programs (*Chen et al., 1997*).

## Brief pauses and complete stops have distinct effects on beta oscillations

We did not find evidence of a beta rebound following the short pause of movement during reversals in motor cortex. Instead, we observed a transient broadband beta power suppression in cortex, which was likely related to re-acceleration in the opposite direction. In contrast, the STN exhibited increases of high beta power in some patients, compatible with post-processing of the brief pause of movement occurring during reversals. On an observational level, the spectral patterns of these increases did not entirely match the individual stop-related beta pattern, lacking the low-beta component of the beta rebound. Thus, STN low-beta oscillations might not re-emerge when stopping briefly within a movement sequence, corroborating a dissociation of low- and high-beta oscillations, as proposed previously (*Chandrasekaran et al., 2019*; *Oswal et al., 2021*; *Patai et al., 2022*). Given that the beta rebound has been reported to slow reaction times (*Muralidharan and Aron, 2021*) and to reduce corticospinal excitability (*Wessel et al., 2016*), and that beta power must decrease for movement to start (*Heinrichs-Graham and Wilson, 2016*; *Khanna and Carmena, 2017*), it is likely that at least the low-beta portion of the beta rebound needs to be avoided during changes of ongoing action because it would slow down re-acceleration otherwise.

In agreement with the current findings, previous research assessing STN- and cortical beta activity reported no beta rebound around the time a movement changed (*Alegre et al., 2004*; *London et al., 2021*), except for one study, which did report a cortical beta rebound between two successive movements (*Muralidharan and Aron, 2021*). It should be noted, however, that the pauses were ~1–2 s long. In our study, the beta rebound occurred only at the end of the movement sequence, when patients were already in the process of stopping and movement had already slowed. This picture emerged irrespective of whether power dynamics were analyzed in a movement- or cue-aligned fashion (see *Figure 3—figure supplement 1*, *Figure 4—figure supplement 1*, *Figure 6—figure supplement 1*). A causal role of the beta rebound in stopping is, therefore, implausible. More likely, the rebound serves as a post-movement feedback signal reflecting task-dependent contextual information used to either confirm or update motor plans (*Alegre et al., 2004*; *Cao and Hu, 2016*). Alternatively, it might indicate the clear-out of the entire motor program (*Schmidt et al., 2019*).

With respect to gamma activity, we observed increases in power at movement start in the contralateral STN and decreases in power at movement termination in contralateral STN and M1. While movement-related increases in gamma power are an established finding in the literature (*Litvak et al., 2012*; *Lofredi et al., 2018*), there appears to be no consensus on its functional role during movement stopping. Previous studies using auditory stop signals reported STN gamma power increases in response to stop signals (*Fischer et al., 2017*; *Ray et al., 2012*). When assessed within a brief critical window between the stop signal and the average time of the upcoming finger tap, gamma power even correlated with stopping success, i.e., gamma was stronger when the downward movement was stopped earlier (*Fischer et al., 2017*). Conversely, another study using visual stop signals reported decreased STN gamma power (*Alegre et al., 2013*). We are unaware of studies that have assessed gamma power changes when stopping a continuous movement in response to visual cues and, therefore, provide first evidence for a decrease in this scenario, although we cannot rule out that focusing on different DBS contacts or using auditory stop signals and shorter event-locked analysis windows might produce different results.

## Limitations and future directions

Invasive measurements of STN activity are only possible in patients who are undergoing or have undergone brain surgery. Studies drawing from this limited pool of candidate participants are typically limited in terms of sample size and cohort stratification, particularly when carried out in a peri-operative setting. Here, we had a sample size of 20, which is rather high for a peri-operative MEG-LFP study, but still low in terms of absolute numbers.

We further acknowledge that most of our participants were older than 60 y. To diminish any confounding effects of age on movement-related modulations of neural oscillations, such as beta suppression and rebound (*Bardouille and Bailey, 2019*; *Espenhahn et al., 2019*), we included age as a covariate in the statistical analyses.

Furthermore, we cannot be sure to what extent the present study's findings relate to PD pathology rather than general motor processing. We suggest that our approach at least approximates healthy

brain functioning as patients were on their usual dopaminergic medication. Dopaminergic medication has been demonstrated to normalize power within the STN and globus pallidus internus, as well as STN-globus pallidus internus and STN-cortex coherence (*Brown et al., 2001*; *Hirschmann et al., 2013*). Additionally, several of our findings match observations made in other patient populations and in healthy participants, who exhibit the same beta power dynamics at movement start and stop (*Alegre et al., 2004*) that we observed here. Notably, our finding of enhanced cortical involvement in face of uncertainty aligns well with established theories of cognitive processing, given the cortex' prominent role in managing higher cognitive functions (*Altamura et al., 2010*). Yet, transferring our approach and task to patients with different disorders, e.g., obsessive compulsive disorder, or examining young and healthy participants solely at the cortical level, could contribute to elucidating whether the synchronization dynamics reported here are indeed independent of PD and age. Additionally, future research could capitalize on sensing-capable devices to circumvent the necessity to record brain activity peri-operatively, allowing for larger sample sizes and to circumvent the stun effect, an immediate improvement in motor symptoms arising as a consequence of electrode implantation (*Mann et al., 2009*). Lastly, given the present study's focus on understanding movement-related rhythms, particularly in the beta range, future research could further explore the role of gamma oscillations in continuous movement and their relation to action potentials in motor areas (*Fischer et al., 2020*; *Igarashi et al., 2013*), which form the basis of movement encoding in the brain.

Due to the diversity of modulations across patients, we cannot provide a general description of how the STN responds to reversals. The variability may result from the fact that the exact recording site varied across patients, although all recording contacts were located in the dorsolateral STN. Furthermore, stop processes, mediated by the hyperdirect and the indirect pathway as well as cortico-striatal go processes, may emerge in the basal ganglia close in time (*Muralidharan et al., 2022*;

**Table 1.** Patient clinical characteristics.

Disease duration refers to the time since diagnosis. For patient 4, the time since first symptom manifestation is given. MoCa = Montreal Cognitive Assessment Test.

| ID | Age | sex | Pre-surgical MDS-UPDRS III ON | Pre-surgical MoCa | Used hand | Disease duration (y) | Motor subtype | DBS Lead |
|---|---|---|---|---|---|---|---|---|
| 1 | 70 | m | 20 | 27 | R | 3 | tremor | Abbott Infinity |
| 2 | 67 | m | 31 | 27 | R | 32 | mixed | Abbott Infinity |
| 3 | 64 | M | 23 | 25 | R | 6 | akinetic-rigid | Abbott Infinity |
| 4 | 57 | M | 53 | 27 | L | 2 | tremor | Abbott Infinity |
| 5 | 66 | F | 33 | 26 | R | 18 | mixed | Abbott Infinity |
| 6 | 75 | M | 11 | 24 | R | 13 | akinetic-rigid | Abbott Infinity |
| 7 | 66 | M | 10 | 27 | R | 13 | mixed | Abbott Infinity |
| 8 | 83 | F | 7 | 25 | R | 13 | tremor | Abbott Infinity |
| 9 | 68 | F | 15 | 20 | L | 11 | akinetic-rigid | Abbott Infinity |
| 10 | 58 | F | 21 | 14 | R | 5 | mixed | Medtronic |
| 11 | 69 | M | 17 | 27 | L | 12 | mixed | Abbott Infinity |
| 12 | 73 | F | 17 | 28 | L | 9 | mixed | Abbott Infinity |
| 13 | 65 | M | 28 | 21 | L | 13 | mixed | Abbott Infinity |
| 14 | 65 | M | 12 | 20 | R | 4 | tremor | Abbott Infinity |
| 15 | 64 | M | 25 | 23 | R | 17 | mixed | Medtronic |
| 16 | 68 | M | 15 | 28 | R | 5 | mixed | Abbott Infinity |
| 17 | 65 | M | 9 | 18 | R | 4 | akinetic-rigid | Abbott Infinity |
| 18 | 50 | M | 11 | 26 | R | 4 | tremor | Abbott Infinity |
| 19 | 68 | F | 42 | 23 | L | 12 | mixed | Abbott Infinity |
| 20 | 56 | F | 20 | 26 | R | 3 | tremor | Medtronic |

*Schmidt and Berke, 2017*), potentially overlapping. The sub-populations processing these signals in the STN (*Isoda and Hikosaka, 2008*; *Schmidt and Berke, 2017*; *Schmidt et al., 2013*) might not be resolvable with macro-electrode LFP recordings.

## Conclusion

In conclusion, we have revealed distinct local and long-range synchronization dynamics of motor cortex and STN during changes of ongoing action in different movement contexts. We found that stopping briefly in the course of changing movement direction and terminating a movement sequence have distinct oscillatory profiles. Moreover, movement scenarios that do not permit movement preparation and require higher levels of caution appear to involve enhanced levels of subthalamo-cortical beta synchronization, highlighting that long-range beta coherence plays an important role in coordinating movements in response to unpredictable events.

# Materials and methods

## Patients

23 PD patients with a mean age of 66.13 y (±7.72 y) participated in the study (*Table 1*). DBS surgery was performed by the Department of Functional Neurosurgery and Stereotaxy of the University Hospital Düsseldorf under full anesthesia and according to standard procedures. 21 patients were implanted with Abbott Infinity segmented leads (Abbott Laboratories, Chicago, Illinois, USA) and three patients with Medtronic SenSight electrodes (Medtronic Inc, Minneapolis, MN, USA). DBS surgery was performed in two steps, and the measurements took place in between the implantation of the electrodes and the implantation of the pulse generator. Prior to participating, all patients provided their written informed consent in agreement with the Declaration of Helsinki. The study was approved by the Ethics Committee of the Medical Faculty of Heinrich Heine University Düsseldorf. The medication schedule was not changed for this experiment (Med ON state). Three participants were excluded from the analyses, two of whom were physically unable to perform the paradigm. The data of the third patient were contaminated by excessive artifacts.

## Recordings

Measurements took place the day after the implantation of DBS electrodes. Externalization of leads allowed us to measure LFPs from the STN in combination with MEG. For LFP recordings, we used a mastoid reference and re-referenced the signals using a bipolar montage post-measurement. MEG signals were acquired simultaneously, using a 306-channel whole-head MEG system (VectorView, MEGIN). Muscle and ocular activity were monitored via electromyography (EMG) and vertical and horizontal electrooculography (EOG), respectively. EMG surface electrodes were placed on patients' right and left forearms, referenced to the muscle tendons at the wrist. We first recorded 5 min of resting-state data, followed by the motor task, which lasted for about 32 min in total. During the task, patients were required to turn a wheel clockwise or counterclockwise, according to visual instructions presented on a screen in front of them.

## Experimental design

Patients were seated in the MEG scanner in a magnetically shielded room with a turning device ('wheel,' *Figure 2b*) placed on a table in front of them. The wheel (diameter = 14 cm, height = 6.5 cm) could be turned into both directions and had indentations, allowing comfortable placement of one index finger for turning. An MEG-compatible plastic fiber optic position sensor system (MR430 Series ZapFREE Fiber Optic Absolute Encoder System, MICRONER Inc, Camarillo, CA, USA) was used to measure wheel turning. The absolute angular position was continuously measured and updated at a frequency of 1.2 kHz. Given its design, the sensing system did not introduce any magnetic interference.

Movement prompts were presented on a screen. The visual stimuli consisted of two curved arrows pointing either clockwise or counterclockwise, respectively, and a stop sign with white font on a red background (*Figure 1a*). Patients were instructed to turn the wheel with their index finger following the direction of the arrows and to stop when a stop cue appeared. We did not impose requirements on turning speed or body side, so that patients could use their less affected hand and adjust the speed to their individual motor capabilities.

The experiment was conducted in two distinct blocks, where stimuli differed in their order and timing of presentation. In the predictable condition, trials consisted of a blue arrow cueing the patients to start turning clockwise, followed by a cue to change the turning direction after 4 s, and a stop cue after another 4 s. Each trial was followed by a pause lasting for 4 s. The condition was termed *predictable*, as the fixed timing and the fixed order of cues allowed patients to easily predict and prepare what they had to do next and when. In the unpredictable condition, the start cue was either clockwise or counterclockwise and was followed by 0, 1, or 2 reversals before the stop cue appeared. Each alternative occurred equally often (go, stop: 33%; go, reverse, stop: 33%; go, reverse, reverse, stop: 33%; clockwise and counterclockwise start directions were balanced). Additionally, the intervals between the visual stimuli were unpredictable (ranging between 4–7 s), with 50% of all inter-stimulus intervals kept at 4 s, as in the predictable condition. Hence, patients could not foresee the sequence and the timing of instructions, calling for a more cautious/attentive monitoring of cues.

We recorded two blocks per condition, with 36 trials each, in a pseudo-randomized fashion. To enhance compliance, we split each block in half, allowing for a short break, and also offered breaks between blocks.

## Materials availability statement

The code used for analyses is available at https://github.com/luciewinkler/Subthalamo-Cortical-Synchronization (copy archived at *Hirschmann and Winkler, 2025*).

## Data analysis

Data were analyzed using MATLAB R2019b (The Mathworks, Natick, Massachusetts, USA) and the toolbox FieldTrip (*Oostenveld et al., 2011*). For statistical testing, we used IBM SPSS Statistics 28 (IBM Corporation, Somers, USA).

## Preprocessing

The data were visually inspected to identify and tag noisy channels and subsequently cleaned using temporal Signal Space Separation to remove artifacts originating from outside the MEG sensor array (*Taulu and Simola, 2006*). Then, the data were downsampled to 500 Hz. We applied a high-pass finite impulse response filter with a cut-off frequency of 1 Hz to remove low-frequency drifts and screened the data for remaining artifacts.

We used custom MATLAB scripts for semi-automated detection of movement start, reversal, and stop in the wheel data. This was achieved by applying an event-specific combination of amplitude and duration thresholds to the first temporal derivative of the rotation angle measurements. To ensure that events were correctly marked, all events were visually inspected and manually corrected if needed. Then, we epoched MEG and LFP data with respect to the behavioral events. Trials were centered around movement events of interest, i.e., start, stop, and reversal of movement, and encompassed 4 s. Movement-aligned angular speed was calculated within those time windows and averaged over trials. Reaction times to cues were defined as the time from cue presentation until movement.

## LFP Channel Selection

The positions of DBS electrodes were localized with the MATLAB toolbox Lead-DBS (*Neudorfer et al., 2023*) using the patients' pre-operative T1- and T2-weighted MRIs (Magnetom Trio MRI scanner, Siemens, Erlangen, Germany) and postoperative CT scans (*Figure 1b*). In order to select one LFP channel for each patient and hemisphere, we identified the bipolar LFP channel with the strongest beta suppression and beta rebound, as previous research has demonstrated the presence of these modulations in the dorsolateral motor STN (*Benis et al., 2014*; *Wessel et al., 2016*). Moreover, the source of subthalamic beta oscillations has been localized to the dorsal STN (*Tamir et al., 2020*).

## Regions of interest

Similarly, we selected cortical regions of interest (ROIs) by localizing the strongest event-related modulations of beta power/beta coherence. For source localization, we first co-registered the pre-operative T1-weighted MRI scans to the MEG coordinate system. Using the segmented MRIs, we prepared forward models based on single-shell realistic head models (*Nolte, 2003*). Beamformer grids, specifying the position of sources, covered the entire brain and were aligned to the Montreal Neurological

Institute (MNI) space. Subsequently, we applied *Dynamic Imaging of Coherent Sources (DICS)* (*Gross et al., 2001*) to beta-band LFP-MEG cross-spectral densities pooled across predictability conditions. Next, we computed contrasts between post-event (0–2 s) and pre-event (−2–0 s) beta power/coherence and averaged the absolute changes across patients and events (movement start, reversal, and stop). This served to identify the regions with the strongest change in general, irrespective of sign, event, and predictability condition.

The strongest beta power modulations localized to the hand knob area of primary motor cortex (M1; *Figure 1a*) and the strongest changes in STN-cortex beta coherence to medial sensorimotor cortex (MSMC; *Figure 1b*). Thus, we focused our analysis on bilateral STN, M1, and MSMC. For time-frequency analysis, we represented each cortical ROI by the grid point of strongest modulation and its six nearest neighbors and extracted a time-series for each grid point, using a linearly constrained minimum variance spatial filter (*Van Veen and Buckley, 1988*).

## Time-frequency analyses

While our main focus was the beta band, we also included other frequencies in our time-frequency analyses to get a more complete picture of power and coherence changes. Specifically, we considered the frequency ranges 5–45 Hz and 55–90 Hz, omitting the 50 Hz line noise artifact, and the time range from −1.6–1.6 s with respect to the movement event. Fourier spectra were computed using a multi-taper approach (four Slepian tapers for the low frequency range and seven Slepian tapers for the high frequency range), a window size of 800 ms and a step size of 50 ms. Using the Fourier coefficients, we computed power and STN-cortex across-trial coherence for each time-frequency bin.

For illustration, we applied baseline correction, using the mean of the pre-event time window (−1.6–0 s) as baseline. In case of power, we expressed changes with respect to baseline in decibel. In case of coherence, we subtracted the baseline values. Time-frequency spectra of cortical sources were averaged over neighboring grid points belonging to the same ROI.

## Granger causality analysis

We computed beta and gamma band non-parametric Granger causality (*Dhamala et al., 2008*) between cortical ROIs and the STN for the post-event time windows (0–2 s with respect to start, reversal, and stop). Because estimates of Granger causality are often biased, we compared the original data to time-reversed data to suppress non-causal interactions. True directional influence is reflected by a higher causality measure in the original data than in its time-reversed version, resulting in a positive difference between the two, the opposite being the case for a signal that is 'Granger-caused' by the other. Directionality is thus reflected by the sign of the estimate (*Haufe et al., 2013*). Because rmANCOVA results indicated no significant effects for predictability and movement type, and post-hoc tests did not show significant differences between hemispheres, we averaged Granger causality estimates over movement types, hemispheres, and predictability conditions in *Figure 6—figure supplement 2*.

## Statistical analysis

Repeated measures analyses of (co)variance (rmANCOVA), implemented in SPSS, were our main tool for statistical analysis. This approach provides a comprehensive, multi-factorial analysis, but requires pre-selection of brain areas (see Regions of Interest), a frequency range, and a time range of interest (see Power and coherence). The dependent variable was either the event-related change in power or coherence, the hemispheric lateralization of the event-related power change (see Lateralization), or Granger causality. The main independent variable of interest was *predictability. Brain area* and *movement type* were also included as factors due to their clear effects on power and coherence, but their main effects are not reported in the main paper. They can be found in the Supplementary material. Because Mauchly's test indicated violations of the sphericity assumption, we report results from the multivariate test (*Rasch et al., 2021*). To account for their potential influence on brain activity, we added age, pre-operative UPDRS score, and disease duration as covariates to all ANOVAs. Covariates were standardized by means of z-scoring.

The rmANCOVAs were complemented by cluster-based permutation tests for detecting significant power/coherence modulations relative to baseline. These tests are mono-factorial but have the advantage of not requiring any preselection of time or frequency ranges while providing correction

for multiple comparisons. The cluster-defining and the cluster significance threshold was set to 0.05 (two-sided test). The cluster statistic was the sum of $t$-values within a cluster. We performed 1000 permutations per test.

### Power and connectivity

To assess the effect of predictability on power, we conducted a repeated measures ANCOVA testing the influence of the factors *movement* (start, stop, reversal), *predictability* (predictable, unpredictable) and *brain area* (STN, M1, MSMC, ipsilateral, and contralateral to the moving hand), as well as interactions between these factors, on the event-related modulation of beta power. Here, modulation refers to the difference between post-event (0–1.6 s) and pre-event (−1.6–0 s) beta power in decibel (dB). A similar rmANCOVA was computed for event-related modulations of STN-cortex beta coherence. In this case, we considered the difference between pre- and post-event coherence. Here, the factor *brain area* contained of the following pairs: contralateral M1-contralateral STN, contralateral MSMC-contralateral STN, ipsilateral M1-ipsilateral STN, ipsilateral MSMC-ipsilateral STN, with the terms ipsilateral and contralateral referring to the moving hand. In an additional rmANCOVA, we considered post-event (0–2 s) Granger causality. The factor *brain area* included these pairs: contralateral M1->contralateral STN, contralateral STN->contralateral M1, contralateral MSMC->contralateral STN, contralateral STN->contralateral MSMC, ipsilateral M1->ipsilateral STN, ipsilateral STN->ipsilateral M1, ipsilateral MSMC-ipsilateral STN, ipsilateral STN->ipsilateral MSMC. This rmANCOVA was supplemented by $t$-tests assessing whether the difference in Granger causality between the reversed and the original data differed from zero, indicating significant directionality.

Because beta power is known to correlate with movement speed (*Lisi and Morimoto, 2015*; *Lofredi et al., 2023*; *Pogosyan et al., 2009*), we added standardized turning speed, averaged over trials and timepoints, as an additional covariate to the above-mentioned rmANCOVAS.

### Lateralization

We compared the beta power suppression and the beta power rebound with respect to their hemispheric lateralization, using a rmANOVA with the factors *brain area* (STN, M1, MSMC), *predictability* (predictable, unpredictable), and *modulation type* (beta suppression, beta rebound). Lateralization was quantified by the lateralization index, defined as the difference between contralateral and ipsilateral power, normalized by power summed over both hemispheres.

### Behavior

To assess whether the predictability of movement prompts had an effect on the patients' performance in the task, we performed a rmANOVA with the factors *predictability* (predictable, unpredictable), *movement* (start, stop, reversal) and their interaction on reaction times and movement-aligned wheel turning speed, averaged over trials and time points, respectively. Epochs without movement (pre-start and post-stop) were disregarded in this analysis.

## Acknowledgements

This research was funded by the Brunhilde Moll Stiftung. The authors thank all participants for their time, cooperation, and willingness to participate. Furthermore, the authors thank Hannah Feldmann for her contributions to developing the paradigm, Dafina Sylaj for her help with patient recruitment, and Lilli Ahrenberg for her work localizing the patients' DBS electrodes.

## Additional information

### Funding

| Funder | Grant reference number | Author |
|---|---|---|
| Brunhilde Moll Stiftung | | Alfons Schnitzler<br>Jan Hirschmann |

| Funder | Grant reference number | Author |
|---|---|---|

The funders had no role in study design, data collection and interpretation, or the decision to submit the work for publication.

## Author contributions

Lucie Winkler, Formal analysis, Validation, Investigation, Visualization, Writing - original draft; Markus Butz, Investigation, Writing – review and editing; Abhinav Sharma, Methodology, Writing – review and editing; Jan Vesper, Resources, Writing – review and editing; Alfons Schnitzler, Resources, Funding acquisition, Writing – review and editing; Petra Fischer, Conceptualization, Resources, Supervision, Methodology, Writing – review and editing; Jan Hirschmann, Conceptualization, Supervision, Funding acquisition, Investigation, Methodology, Project administration, Writing – review and editing

## Author ORCIDs

Lucie Winkler ![ORCID] https://orcid.org/0009-0005-8558-9428
Markus Butz ![ORCID] https://orcid.org/0000-0003-1438-5792
Petra Fischer ![ORCID] https://orcid.org/0000-0001-5585-8977
Jan Hirschmann ![ORCID] https://orcid.org/0000-0001-6315-1912

## Ethics

Prior to participating, all patients provided their written informed consent in agreement with the declaration of Helsinki. The study was approved by the Ethics Committee of the Medical Faculty of Heinrich Heine University Düsseldorf (approval identifier: 14-264).

Reviewer #1 (Public review): https://doi.org/10.7554/eLife.101769.3.sa1
Reviewer #2 (Public review): https://doi.org/10.7554/eLife.101769.3.sa2
Reviewer #3 (Public review): https://doi.org/10.7554/eLife.101769.3.sa3
Author response https://doi.org/10.7554/eLife.101769.3.sa4

# Additional files

## Supplementary files

Supplementary file 1. Behavioral effects. (A) Effects of condition (predictable, unpredictable) and movement (start, reverse, stop) on movement-aligned speed, controlling for age, pre-operative UPDRS score, and disease duration. (B) Effects of condition (predictable, unpredictable) and movement (start, reverse, stop) on reaction times to cues, controlling for age, pre-operative UPDRS score, and disease duration.

Supplementary file 2. Effects on lateralization. Effects of modulation type (beta suppression, beta rebound), condition (predictable, unpredictable), and ROI (STN, M1, MSMC) on lateralization index, controlling for age, pre-operative UPDRS score, and disease duration.

Supplementary file 3. Effects on beta power and coherence. (**A**) Effects of condition (predictable, unpredictable), movement (start, reverse, stop), and regions of interest ROI (contralateral and ipsilateral STN, M1, MSMC) on normalized power, controlling for movement speed, age, pre-operative UPDRS score, and disease duration. (**B**) Effects of condition (predictable, unpredictable), movement (start, reverse, stop), and ROI (contralateral STN-M1, contralateral STN-MSMC, ipsilateral STN-M1, ipsilateral STN-MSMC) on coherence modulation, controlling for movement speed, age, pre-operative UPDRS score, and disease duration.

Supplementary file 4. Effects on beta Granger causality. (**A**) Effects of condition (predictable, unpredictable), movement (start, reverse, stop), and regions of interest (ROI) (contralateral and ipsilateral M1->STN, STN->M1, MSMC->STN, STN->MSMC) on Granger causality, controlling for movement speed, age, pre-operative UPDRS score, and disease duration.

Supplementary file 5. Effects on gamma power and coherence. (**A**) Effects of condition (predictable, unpredictable), movement (start, reverse, stop), and regions of interest (ROI) (contralateral and ipsilateral STN, M1, MSMC) on normalized power, controlling for movement speed, age, pre-operative UPDRS score and disease duration. (**B**) Effects of condition (predictable, unpredictable), movement (start, reverse, stop), and ROI (contralateral STN-M1, contralateral STN-MSMC, ipsilateral STN-M1, ipsilateral STN-MSMC) on coherence modulation, controlling for movement speed, age,

pre-operative UPDRS score, and disease duration.

Supplementary file 6. Effects on gamma Granger causality. (**A**) Effects of condition (predictable, unpredictable), movement (start, reverse, stop), and regions of interest (ROI) (contralateral and ipsilateral M1->STN, STN->M1, MSMC->STN, STN->MSMC) on Granger causality, controlling for movement speed, age, pre-operative UPDRS score, and disease duration.

Supplementary file 7. Excel file containing reaction times and movement speed.

Supplementary file 8. Excel file containing beta and gamma power values.

Supplementary file 9. Excel file containing beta lateralization index values.

Supplementary file 10. Excel file containing the beta and gamma coherence values.

Supplementary file 11. Excel file containing the beta and gamma Granger causality values.

MDAR checklist

## Data availability

The data tables that formed the input to the statistical analyses (band-average power and coherence) are provided as Supplementary Excel files. The raw data is not openly available because patients did not consent to data sharing. Researchers interested in accessing others parts of the data which can be completely de-identifed may contact Jan.Hirschmann@uni-duesseldorf.de for help with seeking approval from the Data Protection Office of the University Clinic Düsseldorf.

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
