## [Editor Report · eLife Assessment]

This **valuable** study combined whole-head magnetoencephalography (MEG) and subthalamic (STN) local field potential (LFP) recordings in patients with Parkinson's disease undergoing deep brain stimulation surgery. The paper provides **convincing** evidence that cortical and STN beta oscillations are sensitive to movement context.

---

## [Referee Report · Reviewer #1 (Public review)]

Summary:

Winkler et al. present brain activity patterns related to complex motor behavior by combining whole-head magnetoencephalography (MEG) with subthalamic local field potential (LFP) recordings from people with Parkinson's disease. The motor task involved repetitive circular movements with stops or reversals associated with either predictable or unpredictable cues. Beta and gamma frequency oscillations are described, and the authors found complex interactions between recording sites and task conditions. For example, they observed stronger modulation of connectivity in unpredictable conditions. Moreover, STN power varied across patients during reversals, which differed from stopping movements. The authors conclude that cortex-STN beta modulation is sensitive to movement context, with potential relevance for movement redirection.

Strengths:

This study employs a unique methodology, leveraging the rare opportunity to simultaneously record both invasive and non-invasive brain activity to explore oscillatory networks.

Weaknesses:

It is difficult to interpret the role of the STN in context of reversals, because no consistent activity pattern emerged.

Comments on revisions: The authors have adequately addressed my comments.

---

## [Referee Report · Reviewer #2 (Public review)]

Summary:

This study examines the role of beta oscillations in motor control, particularly during rapid changes in movement direction among patients with Parkinson's disease. The researchers utilized magnetoencephalography (MEG) and local field potential (LFP) recordings from the subthalamic nucleus to investigate variations in beta band activity within the cortex and STN during the initiation, cessation, and reversal of movements, as well as the impact of external cue predictability on these dynamics. The primary finding indicates that beta oscillations more effectively signify the start and end of motor sequences than transitions within those sequences. The article is well-written, clear, and concise.

Strengths:

The use of a continuous motion paradigm with rapid reversals extends the understanding of beta oscillations in motor control beyond simple tasks. It offers a comprehensive perspective on subthalamo-cortical interactions by combining MEG and LFP.

Comments on revisions: I am satisfied with the revisions. I do not have further comments on the revised manuscript.

---

## [Referee Report · Reviewer #3 (Public review)]

Summary:

The study highlights how the initiation, reversal, and cessation of movements are linked to changes in beta synchronization within the basal ganglia-cortex loops. It was observed that different movement phases, such as starting, stopping briefly, and stopping completely, affect beta oscillations in the motor system.

It was found that unpredictable cues lead to stronger changes in STN-cortex beta coherence. Additionally, specific patterns of beta and gamma oscillations related to different movement actions and contexts were observed. Stopping movements was associated with a lack of the expected beta rebound during brief pauses within a movement sequence.

Overall, the results underline the complex and context-dependent nature of motor control and emphasize the role of beta oscillations in managing movement according to changing external cues.

Strengths:

The paper is very well written, clear and appears methodologically sound.

Although the use of continuous movement (turning) with reversals is more naturalistic than many previous button push paradigms.

Weaknesses:

The generalizability of the findings are somewhat curtailed by the fact that this was performed peri-operatively during the period of the microlesion effect. Given the availability of sensing enabled DBS devices now and HD-EEG, does MEG offer a significant enough gain in spatial localizability to offset the fact that it has to be done shortly postoperatively with externalized leads, with attendant stun effect? Specifically, for paradigms that are not asking very spatially localized questions as a primary hypothesis?

Further investigation of the gamma signal seems warranted, even though it has a slightly lower proportional change in amplitude in beta. Given that the changes in gamma here are relatively wide band, this could represent a marker of neural firing that could be interestingly contrasted against the rhythm account presented.

Comments on revisions: I congratulate the authors on their paper and their revisions and I have no further comments. I look forward to seeing the continuous analyses in the future. Good luck!

---

## [Author Response]

The following is the authors’ response to the original reviews.

**eLife Assessment**
This valuable study combined whole-head magnetoencephalography (MEG) and subthalamic (STN) local field potential (LFP) recordings in patients with Parkinson's disease undergoing deep brain stimulation surgery. The paper provides solid evidence that cortical and STN beta oscillations are sensitive to movement context and may play a role in the coordination of movement redirection.

We are grateful for the expert assessment by the editor and the reviewers. Below we provide pointby-point replies to both public and private reviews. We have tried to keep the answers in the public section short and concise, not citing the changed passages unless the point does not re-appear in the recommendations. There, we did include all of the changes to the manuscript, such that the reviewers need not go back and forth between replies and manuscript.

The reviewer comments have not only led to numerous improvements of the text, but also to new analyses, such as Granger causality analysis, and to methodological improvements e.g. including numerous covariates in the statistical analyses. We believe that the article improved substantially through the feedback, and we thank the reviewers and the editor for their effort.

**Public Reviews**

**Reviewer #1 (Public review):**
Summary:Winkler et al. present brain activity patterns related to complex motor behaviour by combining wholehead magnetoencephalography (MEG) with subthalamic local field potential (LFP) recordings from people with Parkinson's disease. The motor task involved repetitive circular movements with stops or reversals associated with either predictable or unpredictable cues. Beta and gamma frequency oscillations are described, and the authors found complex interactions between recording sites and task conditions. For example, they observed stronger modulation of connectivity in unpredictable conditions. Moreover, STN power varied across patients during reversals, which differed from stopping movements. The authors conclude that cortex-STN beta modulation is sensitive to movement context, with potential relevance for movement redirection.Strengths:This study employs a unique methodology, leveraging the rare opportunity to simultaneously record both invasive and non-invasive brain activity to explore oscillatory networks.Weaknesses:It is difficult to interpret the role of the STN in the context of reversals because no consistent activity pattern emerged.

We thank the reviewer for the valuable feedback to our study. We agree that the interpretation of the role of the STN during reversals is rather difficult, because reversal-related STN activity was highly variable across patients. Although there seem to be consistent patterns in sub-groups of the current cohort, with some patients showing event-related increases (Fig. 3b) and others showing decreases, the current dataset is not large enough to substantiate or even explain the existence of such clusters. Thus, we limit ourselves to acknowledging this limitation and discussing potential reasons for the high variability, namely variability in electrode placement and insufficient spatial resolution for the separation of specialized cell ensembles within the STN (see Discussion, section Limitations and future directions).

**Reviewer #2 (Public review):**
Summary:This study examines the role of beta oscillations in motor control, particularly during rapid changes in movement direction among patients with Parkinson's disease. The researchers utilized magnetoencephalography (MEG) and local field potential (LFP) recordings from the subthalamic nucleus to investigate variations in beta band activity within the cortex and STN during the initiation, cessation, and reversal of movements, as well as the impact of external cue predictability on these dynamics. The primary finding indicates that beta oscillations more effectively signify the start and end of motor sequences than transitions within those sequences. The article is well-written, clear, and concise.Strengths:The use of a continuous motion paradigm with rapid reversals extends the understanding of beta oscillations in motor control beyond simple tasks. It offers a comprehensive perspective on subthalamocortical interactions by combining MEG and LFP.Weaknesses:(1) The small and clinically diverse sample size may limit the robustness and generalizability of the findings. Additionally, the limited exploration of causal mechanisms reduces the depth of its conclusions and focusing solely on Parkinson's disease patients might restrict the applicability of the results to broader populations.

We thank the reviewer for the insightful feedback. We address these issues one by one in our responses to points 2, 4 and 6, respectively.

(2) The small sample size and variability in clinical characteristics among patients may limit the robustness of the study's conclusions. It would be beneficial for the authors to acknowledge this limitation and propose strategies for addressing it in future research. Additionally, incorporating patient-specific factors as covariates in the ANOVA could help mitigate the confounding effects of heterogeneity.

Thank you for this comment. The challenges associated with recording brain activity peri-operatively can be a limiting factor when it comes to sample size and cohort stratification. We now acknowledge this in the revised discussion (section Limitations and future directions). Furthermore, we suggest using sensing-capable devices in the future as a measure to increase sample sizes (Discussion, section Limitations and future directions). Lastly, we appreciate the idea of adding patient-specific factors as covariates to the ANOVAs and have thus included age, disease duration and pre-surgical UPDRS score into our models. This did not lead to any qualitative changes of statistical effects.

(3) The author may consider using standardized statistics, such as effect size, that would provide a clearer picture of the observed effect magnitude and improve comparability.

Thanks for the suggestion. As measures of effect size, we have added partial eta squared (*ηp<sup2*) to the results of all ANOVAs and Cohen’s d to all follow-up *t*-tests.

(4) Although the study identifies relevance between beta activity and motor events, it lacks causal analysis and discussion of potential causal mechanisms. Given the valuable datasets collected, exploring or discussing causal mechanisms would enhance the depth of the study.

We appreciate this idea and have conducted Granger causality analyses in response to this comment. This new analysis reveals that there is a strong cortical drive to the STN for all movements of interest and predictability conditions in the beta band. The detailed results can be viewed on p. 16 in the section on Granger causality. For statistical testing, we conducted an rmANCOVA, similar to those for power and coherence (see p. 46-48 and 54-56 for the corresponding tables), as well as t-tests assessing directionality (Figure 6-figure supplement 2 on p. 35). In the discussion section, we connect these results with prior findings suggesting that the frontal cortex drives the STN in the beta band, likely through hyperdirect pathway fibers (p. 17).

(5) The study cohort focused on senior adults, who may exhibit age-related cortical responses during movement planning in neural mechanisms. These aspects were not discussed in the study.

We appreciate the comment and agree that age may have impacted neural oscillatory activity of patients in the present study. We now acknowledge this in the limitations section, and point out that our approach to handling these effects was including age as a covariate in the statistical analyses.

(6) Including a control group of patients with other movement disorders who also undergo DBS surgery would be beneficial. Because we cannot exclude the possibility that the observed findings are specific to PD or can be generalized. Additionally, the current title and the article, which are oriented toward understanding human motor control, may not be appropriate.

We thank the reviewer for this comment and fully agree that it cannot be ruled out that the present findings are, in part, specific to PD. We acknowledge this limitation in the Limitations and future directions section (p. 20-21). Indeed, including a control group of patients with other disorders would be ideal, but the scarcity of patients with diseases other than PD who receive STN DBS in our centre makes this an unfeasible option in practical terms. We do suggest that future research may address this issue by extending our approach to different disorders or healthy participants on the cortical level (p. 21). Lastly, we appreciate the idea to adjust the title of the present article. The adjusted title is: “Context-Dependent Modulations of Subthalamo-Cortical Synchronization during Rapid Reversals of Movement Direction in Parkinson’s Disease”.

That being said, we do believe that our findings at least approximate healthy functioning and are not solely related to PD. For one, patients were on their usual dopaminergic medication and dopamine has been found to normalize pathological alterations of beta activity. Further, the general pattern of movement-related beta and gamma oscillations reported here has been observed in numerous diseases and brain structures, including cortical beta oscillations measured non-invasively in healthy participants.

**Reviewer #3 (Public review):**
Summary:The study highlights how the initiation, reversal, and cessation of movements are linked to changes in beta synchronization within the basal ganglia-cortex loops. It was observed that different movement phases, such as starting, stopping briefly, and stopping completely, affect beta oscillations in the motor system.It was found that unpredictable cues lead to stronger changes in STN-cortex beta coherence. Additionally, specific patterns of beta and gamma oscillations related to different movement actions and contexts were observed. Stopping movements was associated with a lack of the expected beta rebound during brief pauses within a movement sequence.Overall, the results underline the complex and context-dependent nature of motor-control and emphasize the role of beta oscillations in managing movement according to changing external cues.Strengths:The paper is very well written, clear, and appears methodologically sound.Although the use of continuous movement (turning) with reversals is more naturalistic than many previous button push paradigms.Weaknesses:The generalizability of the findings is somewhat curtailed by the fact that this was performed perioperatively during the period of the microlesion effect. Given the availability of sensing-enabled DBS devices now and HD-EEG, does MEG offer a significant enough gain in spatial localizability to offset the fact that it has to be done shortly postoperatively with externalized leads, with an attendant stun effect? Specifically, for paradigms that are not asking very spatially localized questions as a primary hypothesis?

We appreciate the reviewer’s feedback and acknowledge the valid point raised on the timing of our measurements. Indeed, sensing-enabled devices offer a valid alternative to peri-operative recordings, circumventing the stun effect. We acknowledge this in the revised discussion, section Limitations and future directions (p. 23): “Additionally, future research could capitalize on sensingcapable devices to circumvent the necessity to record brain activity peri-operatively, facilitating larger sample sizes and circumventing the stun effect, an immediate improvement in motor symptoms arising as a consequence of electrode implantation (Mann et al., 2009).” This alternative strategy, however, was not an option here because we did not have a sufficient number of patients implanted with sensing-enabled devices at the time when the data collection was initialized.

That being said, we would like to highlight that in the present study, our goal was not to study pathology related to Parkinson’s disease. Rather, we aimed to learn about motor control in general. The stun effect may have facilitated motor performance in our patients, which is actually beneficial to the research goals at hand.

Further investigation of the gamma signal seems warranted, even though it has a slightly lower proportional change in amplitude in beta. Given that the changes in gamma here are relatively wide band, this could represent a marker of neural firing that could be interestingly contrasted against the rhythm account presented.

We appreciate the reviewer’s interest and we have extended the investigation of gamma oscillations. We now provide statistics regarding the influence of predictability on gamma power and gamma coherence (no significant effects) and explore Granger causality in the gamma (and beta) band (see comment 4 of reviewer 2). Unfortunately, we cannot measure spiking via the DBS electrode, and therefore we cannot investigate correlations between gamma oscillatory activity and action potentials. We do agree with the reviewer, however, that action potentials rather than oscillations form the basis of motor control in the brain. This view of ours is now reflected in the revised discussion, section Limitations and future directions (p. 21): “Lastly, given the present study’s focus on understanding movement-related rhythms, particularly in the beta range, future research could further explore the role of gamma oscillations in continuous movement and their relation to action potentials in motor areas (Fischer et al., 2020; Igarashi, Isomura, Arai, Harukuni, & Fukai, 2013), which form the basis of movement encoding in the brain.”

**Recommendations for the authors:**

**Reviewer #1 (Recommendations for the authors):**
This is a well-conducted study and overall the results are clear. I only have one minor suggestion for improvement of the manuscript. I found the order of appearance of the results somewhat confusing, switching from predictability-related behavioral effects to primarily stopping and reversal-related neurophysiological effects, back to predictability but starting with coherence. I would suggest that the authors try to follow a systematic order focused on the questions at hand. E.g. perhaps readability could be improved if the results section is split into reversal vs. stopping related effects, reporting behavior, power, and coherence in this order, followed by a predictability section, again reporting behavior, power, and coherence. Obviously, this is an optional suggestion. Apart from that, I just missed a more direct message related to the absence of statistical significance related to STN power changes during reversal. I think this could be made more clear in the text.

We thank the reviewer for the feedback to our study. In order to ease reading, we modified the order and further added additional sub-titles to the results section. We start with Behavior (p. 4) and then move on to Power (general movement effects on power – movement effects on STN power – movement effects on cortical power – predictability effects on power). Next, we move on to Connectivity (movement effects on connectivity – predictability effects on connectivity – Granger causality). We hope that these adaptations will help guide the reader.

Additionally, we thank the reviewer for noting that we did not explicitly mention the lack of statistical significance of reversal-related beta power modulations in the STN. We have adapted the section on modulation of STN beta power associated with reversals (p. 8) to: “In the STN, reversals were associated with a brief modulation of beta power, which was weak in the group-average spectrum and did not reach significance (Fig. 3a).”

**Reviewer #2 (Recommendations for the authors):**
(1) The small sample size and variability in clinical characteristics among patients may limit the robustness of the study's conclusions. It would be beneficial for the authors to acknowledge this limitation and propose strategies for addressing it in future research. Additionally, incorporating patient-specific factors as covariates in the ANOVA could help mitigate the confounding effects of heterogeneity.

Thank you for this comment. The challenges associated with recording brain activity peri-operatively can be a limiting factor when it comes to sample size. We now acknowledge this in the revised discussion, section Limitations and future directions (p. 20):

“Invasive measurements of STN activity are only possible in patients who are undergoing or have undergone brain surgery. Studies drawing from this limited pool of candidate participants are typically limited in terms of sample size and cohort stratification, particularly when carried out in a peri-operative setting. Here, we had a sample size of 20, which is rather high for a peri-operative study, but still low in terms of absolute numbers.”

Furthermore, we suggest using sensing-capable devices in the future as a measure to increase sample sizes (p. 21):

“Additionally, future research could capitalize on sensing-capable devices to circumvent the necessity to record brain activity peri-operatively, facilitating larger sample sizes and circumventing the stun effect, an immediate improvement in motor symptoms arising as a consequence of electrode implantation (Mann et al., 2009).”

Lastly, we appreciate the idea of adding patient-specific factors as covariates to the ANOVAs and have thus included age, disease duration and pre-surgical UPDRS score into our models. This did not lead to any qualitative changes of statistical effects.

Revised article

Methods, Statistical analysis:

“To account for their potential influence on brain activity, we added age, pre-operative UPDRS score, and disease duration as covariates to all ANOVAs. Covariates were standardized by means of zscoring.”

(2) The author may consider using standardized statistics, such as effect size, that would provide a clearer picture of the observed effect magnitude and improve comparability.

Thanks for this useful suggestion. As measures of effect size, we have added partial eta squared (*ηp<sup2*) to the results of all ANOVAs and Cohen’s d to all follow-up _t-_tests.

(3) Although the study identifies relevance between beta activity and motor events, it lacks causal analysis and discussion of potential causal mechanisms. Given the valuable datasets collected, exploring or discussing causal mechanisms would enhance the depth of the study.

We appreciate this idea and have conducted Granger causality analyses in response to this comment. This new analysis reveals that there is a strong cortical drive to the STN for all movements of interest and predictability conditions in the beta band, but no directed interactions in the gamma band. For statistical testing, we conducted an rmANCOVA, similar to the analysis of power and coherence (see p. 46-48 and 54-56 for the corresponding tables), as well as *t*-tests assessing directionality (Figure 6 figure supplement 2 on p. 35). In the discussion section, we connect these results with prior findings suggesting that the frontal cortex drives the STN in the beta band, likely through hyperdirect pathway fibers (p. 17).

Revised article

Methods Section, Granger Causality Analysis

“We computed beta and gamma band non-parametric Granger causality (Dhamala, Rangarajan, & Ding, 2008) between cortical ROIs and the STN in the hemisphere contralateral to movement for the post-event time windows (0 – 2 s with respect to start, reversal, and stop). Because estimates of Granger causality are often biased, we compared the original data to time-reversed data to suppress non-causal interactions. True directional influence is reflected by a higher causality measure in the original data than in its time-reversed version, resulting in a positive difference between the two, the opposite being the case for a signal that is “Granger-caused” by the other. Directionality is thus reflected by the sign of the estimate (Haufe, Nikulin, Müller, & Nolte, 2013). Because rmANCOVA results indicated no significant effects for predictability and movement type, and post-hoc tests did not detect significant differences between hemispheres, we averaged Granger causality estimates over movement types, hemispheres and predictability conditions in Figure 6-figure supplement 2.”

Results, Granger causality

“In general, cortex appeared to drive the STN in the beta band, regardless of the movement type and predictability condition. This was reflected in a main effect of ROI on Granger causality estimates (*FROI*(7,9) = 3.443, *pROI* = 0.044, *ηp<sup2* = 0.728; refer to Supplementary File 4 for the full results of the ANOVA). In the hemisphere contralateral to movement, follow-up *t*-tests revealed significantly higher Granger causality estimates from M1 to the STN (*t* = 3.609, one-sided *p* < 0.001, *d* = 0.807) and from MSMC to the STN (*t* = 2.051, one-sided *p* < 0.027, *d* = 0.459) than the other way around. The same picture emerged in the hemisphere ipsilateral to movement (M1 to STN: *t* = 3.082, one-sided *p* = 0.003, *d* = 0.689; MSMC to STN: *t* = 1.833, one-sided *p* < 0.041, *d* = 0.410). In the gamma band, we did not detect a significant drive from one area to the other (*FROI*(7,9) = 0.338, *pROI* = 0.917, *ηp<sup2* = 0.208, Supplementary File 6). Figure 6-figure supplement 2 demonstrates the differences in Granger causality between original and time-reversed data for the beta and gamma band.”

Discussion, The dynamics of STN-cortex coherence

“Considering the timing of the increase observed here, the STN’s role in movement inhibition (Benis et al., 2014; Ray et al., 2012) and the fact that frontal and prefrontal cortical areas are believed to drive subthalamic beta activity via the hyperdirect pathway (Chen et al., 2020; Oswal et al., 2021) it seems plausible that the increase of beta coherence reflects feedback of sensorimotor cortex to the STN in the course of post-movement processing. In line with this idea, we observed a cortical drive of subthalamic activity in the beta band.”

(4) The study cohort focused on senior adults, who may exhibit age-related cortical responses during movement planning in neural mechanisms. These aspects were not discussed in the study.

We appreciate the comment and agree that age may have impacted neural oscillatory activity of patients in the present study. We now acknowledge this in the limitations section, and point out that our approach to handling these effects was including age as a covariate in the statistical analyses.

Revised article

Discussion, Limitations and Future Directions

“Further, most of our participants were older than 60 years. To diminish any confounding effects of age on movement-related modulations of neural oscillations, such as beta suppression and rebound (Bardouille & Bailey, 2019; Espenhahn et al., 2019), we included age as a covariate in the statistical analyses.”

(5) Including a control group of patients with other movement disorders who also undergo DBS surgery would be beneficial. Because we cannot exclude the possibility that the observed findings are specific to PD or can be generalized. Additionally, the current title and the article, which are oriented toward understanding human motor control, may not be appropriate.

We thank the reviewer for this comment and fully agree that it cannot be ruled out that the present findings are, in part, specific to PD. We acknowledge this limitation in the Limitations and future directions section (p. 20-21). Indeed, including a control group of patients with other disorders would be ideal, but the scarcity of patients with diseases other than PD who receive STN DBS makes this an unfeasible option. We do suggest that future research may address this issue by extending our approach to different disorders or healthy participants on the cortical level (p. 21). Lastly, we appreciate the idea to adjust the title of the present article. The adjusted title is: “Context-Dependent Modulations of Subthalamo-Cortical Synchronization during Rapid Reversals of Movement Direction in Parkinson’s Disease”.

That being said, we do believe that our findings at least approximate healthy functioning and are not solely related to PD. For one, patients were on their usual dopaminergic medication for the study and dopamine has been found to normalize pathological alterations of beta activity. More importantly, the general pattern of movement-related beta and gamma oscillations has been observed in numerous diseases and brain structures, including cortical beta oscillations measured non-invasively in healthy participants. Thus, it is not unlikely that the new aspects discovered here are also general features of motor processing.

Revised article

Discussion, Limitations and future directions

“Furthermore, we cannot be sure to what extent the present study’s findings relate to PD pathology rather than general motor processing. We suggest that our approach at least approximates healthy brain functioning as patients were on their usual dopaminergic medication. Dopaminergic medication has been demonstrated to normalize power within the STN and globus pallidus internus, as well as STN-globus pallidus internus and STN-cortex coherence (Brown et al., 2001; Hirschmann et al., 2013). Additionally, several of our findings match observations made in other patient populations and healthy participants, who exhibit the same beta power dynamics at movement start and stop (Alegre et al., 2004) that we observed here. Notably, our finding of enhanced cortical involvement in face of uncertainty aligns well with established theories of cognitive processing, given the cortex' prominent role in managing higher cognitive functions (Altamura et al., 2010). Yet, transferring our approach and task to patients with different disorders, e.g. obsessive compulsive disorder, or examining young and healthy participants solely at the cortical level, could contribute to elucidating whether the synchronization dynamics reported here are indeed independent of PD and age.”

**Reviewer #3 (Recommendations for the authors):**
Despite the strengths of the "rhythm" account of cognitive processes, the paper could possibly be improved by making it less skewed to rhythms explaining all of the movement encoding.

Thank you for this comment - the point is well taken. There is a large body of literature relating neural oscillations to spiking in larger neural populations, which itself is likely the most relevant signal with respect to motor control. In our eyes, it is this link that justifies the rhythm account, i.e. we agree with the reviewer that action potentials are the basis of movement encoding in the brain, not oscillations. Unfortunately, we cannot measure spiking with the method at hand.

To better integrate this view into the current manuscript, we make the following suggestion for future research in the Limitations and future directions section (p. 21): “Lastly, given the present study’s focus on understanding movement-related rhythms, particularly in the beta range, future research could further explore the role of gamma oscillations in continuous movement and their relation to action potentials in motor areas (Fischer et al., 2020; Igarashi, Isomura, Arai, Harukuni, & Fukai, 2013), which form the basis of movement encoding in the brain.”

In Figure 5 - is the legend correct? Is it really just a 0.2% change in power only? That would be a very surprisingly small effect size.

We thank the reviewer for noting this. Indeed, the numbers on the scale quantify relative change (post - pre)/pre and should be multiplied by 100 to obtain %-change. We have adjusted the color bars accordingly.

The dissociation between the effects of unpredictable cues in coherence versus raw power is interesting and could potentially be directly contrasted further in the discussion (here they are presented separately with separate discussions, but this seems like a pretty important and novel finding as beta coherence and power usually go in the same direction).

We appreciate the reviewer’s interest in our findings on the predictability of movement instructions. In case of coherence, the difference between pre- and post-event was generally more positive in the unpredictable condition, meaning that suppressions (negative pre-post difference) were diminished whereas increases (positive pre-post difference) were enhanced. With respect to power, we also observed less suppression in the unpredictable condition at movement start. Therefore, the direction of change is in fact the same. We made this clearer in the revised version by adapting the corresponding sections of the abstract, results and discussion (see below).

The only instance of coherence and power diverging (on a qualitative level) was observed during reversals: here, we noted post-event increases in coherence and post-event decreases in M1 power in the group-average spectra. However, when comparing the pre- and post-event epochs statistically by means of permutation testing, the coherence increase did not reach significance. Hence, we did not highlight this aspect.

Revised version

Abstract

“… Event-related increases of STN-cortex beta coherence were generally stronger in the unpredictable than in the predictable condition. … “

Results, Effects of predictability on beta power

“With respect to the effect of predictability of movement instructions on beta power dynamics (research aim 2), we observed an interaction between movement type and condition (*F*_cond*mov_ (2,14) = 4.206, *p*_cond*mov_ = 0.037, *ηp<sup2* = 0.375), such that the beta power suppression at movement start was generally stronger in the predictable (*M* = -0.170, *SD* = 0.065) than in the unpredictable (*M* = -0.154, *SD* = 0.070) condition across ROIs (*t* = -1.888, one-sided *p* = 0.037, *d* = -0.422). We did not observe any modulation of gamma power by the predictability of movement instructions (*F*_cond_ (1,15) = 0.792, *p*_cond_ = 0.388, *ηp<sup2* = 0.050, Supplementary File 5).”

Effects of predictability on STN-cortex coherence

“With respect to the effect of predictability of movement instructions on beta coherence (research aim 2), we found that the pre-post event differences were generally more positive in the unpredictable condition (main effect of predictability condition; *F*_cond_(1,15) = 8.684, *p*_cond_ = 0.010, *ηp<sup2* = 0.367; Supplementary File 3), meaning that the suppression following movement start was diminished and the increases following stop and reversal were enhanced in the unpredictable condition (Fig. 6a). This effect was most pronounced in the MSMC (Fig. 6b). When comparing regionaverage TFRs between the unpredictable and the predictable condition, we observed a significant difference only for stopping (*t*_clustersum_ = 142.8, *p* = 0.023), suggesting that the predictability effect was mostly carried by increased beta coherence following stops. When repeating the rmANCOVA for preevent coherence, we did not observe an effect of predictability (*F*_cond_(1,15) = 0.163, *p*_cond_ = 0.692, *ηp<sup2* = 0.011), i.e. the effect was most likely not due to a shift of baseline levels. The increased tendency for upward modulations and decreased tendency for downward modulations rather suggests that the inability to predict the next cue prompted intensified event-related interaction between STN and cortex. STN-cortex gamma coherence was not modulated by predictability (*F*_cond_(1,15) = 0.005, *p*_cond_ = 0.944, *ηp<sup2* = 0.000, Supplementary File 5).”

Discussion, Beta coherence and beta power are modulated by predictability

“In the present paradigm, patients were presented with cues that were either temporally predictable or unpredictable. We found that unpredictable movement prompts were associated with stronger upward modulations and weaker downward modulations of STN-cortex beta coherence, likely reflecting the patients adopting a more cautious approach, paying greater attention to instructive cues. Enhanced STN-cortex interactions might thus indicate the recruitment of additional neural resources, which might have allowed patients to maintain the same movement speed in both conditions. […]”

With respect to power, we observed reduced beta suppression in the unpredictable condition at movement start, consistent with the effect on coherence, likely demonstrating a lower level of motor preparation.

Given that you have a nice continuous data task here - the turning of the wheel, it might be interesting to cross-correlate the circular position (and separately - velocity) of the turning with the envelope of the beta signal. This would be a nice finding if you could also show that the beta is modulated continuously by the continuous movements. In the natural world, we rarely do a continuous movement with a sudden reversal, or stop, most of the time we are in continuous movement. Look at this might also be a strength of your dataset.

We could not agree more. In fact, having a continuous behavioral output was a major motivation for choosing this particular task. We are very interested in state space models such as preferential subspace identification (Sani et al., 2021), for example. These models relate continuous brain signals to continuous behavioral target variables and should be of great help for questions such as: do oscillations relate to moment-by-moment adaptations of continuous movement? Which frequency bands and brain areas are important? Is angular position encoded by different brain areas/frequency bands than angular speed? These analyses are in fact ongoing. This project, however, is too large to fit into the current article.